# A Survey of 1000 Respondents on the Polish Population’s Knowledge and Attitudes about Tissue/Organ Donation and Transplantation in Times of Allogeneic Tissue Shortage

**DOI:** 10.3390/ijerph192113875

**Published:** 2022-10-25

**Authors:** Anna Hepa, Wojciech Łabuś, Magdalena Szatan, Marcin Gierek, Artur Kamiński, Karol Szyluk, Paweł Niemiec, Justyna Glik, Diana Kitala

**Affiliations:** 1Dr. Stanislaw Sakiel Burn Treatment Centre in Siemianowice Slaskie, 41-100 Siemianowice Śląskie, Poland; 2Department of Transplantology and Central Tissue Bank, Medical University of Warsaw, 02-091 Warsaw, Poland; 3National Centre for Tissue and Cell Banking, ul. Chałubińskiego 5, 02-004 Warsaw, Poland; 4Department of Physiotherapy, Faculty of Health Sciences in Katowice, Medical University of Silesia in Katowice, 40-752 Katowice, Poland; 5I Department of Orthopaedic and Trauma Surgery, District Hospital of Orthopaedics and Trauma Surgery, 41-940 Piekary Śląskie, Poland; 6Department of Biochemistry and Medical Genetics, Faculty of Health Sciences in Katowice, Medical University of Silesia in Katowice, 40-752 Katowice, Poland; 7Faculty of Health Sciences in Katowice, Medical University of Silesia in Katowice, 40-752 Katowice, Poland; 8Medical Research Agency, 00-014 Warsaw, Poland

**Keywords:** transplantation, skin, donor, attitude, knowledge

## Abstract

Tissue-engineered human allogeneic skin grafts retrieved from a deceased donor play an important role in the therapy of extensive and deeply burned patients. However, there is a vital deficit of allogeneic skin donors, and the reserves of human allogeneic skin grafts are not sufficient. The goal of this work was to analyze the level of knowledge and attitudes of Polish society in the field of transplantation, with particular emphasis on allogeneic skin transplantation. The study used a self-made questionnaire comprised of 23 questions. 1000 respondents took part in this research. The respondents were a diverse group in terms of age, sex, education, and place of residence. The obtained results show a general positive attitude of the respondents towards the idea of transplantology. However, people with lower education presented a more negative attitude towards the donation of tissues and organs. Additionally younger people were not able to clearly declare readiness for organ procurement. What is more data analysis revealed certain gaps in more detailed knowledge and surprising attitudes. In that respect, the lack of awareness about the criteria for determining brain death could be mentioned. There was also a lack of acceptance for skin procurement in specific population groups. It can therefore be concluded that a key role in the success of the idea of transplantation in Poland is the broad and systematic education of the society.

## 1. Introduction

Human allogeneic skin grafts retrieved from a deceased donor play an important role in the therapy of extensive and deeply burned patients [1]. The importance of grafts increases dramatically in the case of casualties of a fire mass catastrophe [2]. However, the effectiveness of human allogeneic skin grafts is limited by the availability of tissue material retrieved from deceased donors. The most important problem to be solved in transplantology is the shortage of donors and thus the shortage of organs and tissues in relation to the number of potential recipients [2,3]. On the other hand, a serious dissonance between the number of multiorgan donations and tissue donations can be seen [3] which worsens when the skin is taken into consideration [2]. Despite the many reports suggesting that the level of awareness about donation and transplantation is satisfactory [4,5,6] the level of actual organ retrievals often remains unsatisfactory [4]. In Poland, a country of approx. 38 million citizens, 529 multi-organ donations (real deceased donors) took place in 2020, while only 54 skin donations took place [7]. The likely cause of this unfavorable phenomenon may be the fact that in Poland there are only two centers for the procurement, preparation and distribution of human allogeneic skin grafts. There are therefore two procurement teams that collect allogeneic skin and, what is more, their range is locally limited. In this approach, there are proposals for systemic solutions which could lead to a central bank of allogeneic skin establishment in the event of a mass disaster (including war) [2]. However, at present, a frequently observed situation is retrieval of tissues/organs from a deceased donor during which, for example, corneas are retrieved, and the skin is not. It should be remarked that the number of corneal retrievals was 744 while skin donation was 54 in 2020 [2,7]. The high number of corneal retrievals may be due to the fact that corneas are procured by more centers and additionally in forensic and funeral homes. Therefore, paradoxically, the greater number of donations of this type may result in greater awareness of the role of corneal transplants as well as the course of the donation itself, which translates into a greater number of donations. Nevertheless, there is concern among the public, and possibly the medical staff, regarding the skin procurement. [data not shown]. In this specific approach, not only the attitudes and knowledge of society should be considered, but also the attitudes of medical personnel. Xie et al. (2017) reported that the attitude towards donation and transplantation in the hospitals is not overly optimistic, and improvement in the training regarding transplantation and donation even among nurses is necessary [8]. Zhang et al. (2017) and Wilczek-Rużyczka et al. (2014) proved that there is a need to provide appropriate training regarding donation to increase donation rates [9,10]. The results of the available studies differ depending on the country and studied group. In order to perform an efficient educational and awareness raising campaign, it is crucial to know the target group, and due to the generation gap, present campaigns may be outdated [11,12]. It has been proved that education sessions are able to address myths about transplantation and help to understand its purposes [12]. Therefore, an increase in the number of donations can be expected. However, it should be remarked that Milaniak et al. reported that in 2016 in Poland 78.5% of Polish respondents agreed to posthumous life-saving organ donation, which is not reflected in the number of skin donations [11].

It should be once more remarked that allogeneic human skin graft therapy in the treatment of burns is an established method of treating burns [1], confirmed by the recommendations of the European Burns Association (EBA). A shortage of allogeneic skin may prove fatal for burns patients, therefore there is an urgent need for preventive actions [2].

The aim of this study was to analyze the state of knowledge and attitudes of the Polish society in the field of transplantology, with particular emphasis on the collection of allogeneic skin from deceased donors.

## 2. Materials and Methods

The study consisted of 23 closed questions and the survey was anonymous. The study lasted from 4 January 2021 to 1 March 2021 to achieve a response rate of 1000 surveys. Both a printed survey and an electronic survey generated by the Google form were used. In the case of the electronic version of the survey, the available online tools for social media (e.g., Facebook, LinkedIn, Sunnyvale, CA, USA) were used to disseminate the survey. The questionnaire was distributed to the network of contacts, but no form of inducement or coercion was used to participate in the study. The recruitment of respondents to the study consisted of voluntary consent to participate in the survey. In case of printed version of the questionnaire participants were recruited among nursing students, patients, surgical clinic patients, medical and administrative staff, employees of selected industrial plants, family members of researchers. This research was fully anonymous.

Excel (Microsoft) was used to work with the data, including database cleaning. Questions and survey are attached in Appendix A.

The proposed research did not need a Bioethics Committee approval, due to the Polish law. It is fully anonymous survey data from the questionnaire.

### Statistical Analysis

All tests were performed with the STATISTICA 12 software (StatSoft, Dell Inc., Round Rock, TX, USA). The distribution was analyzed with the Shapiro-Wilk test and the equality of variance was checked with Levene’s test. Statistical hypothesis testing for two independent samples was determined by the U Mann-Whitney test. For comparing more than two groups of independent samples which did not meet the normality assumption, the Kruskal-Wallis test was used. Chi-square was used for dichotomy data. Pearson’s parametric correlation test and Spearman’s non-parametric test were used. The significance level was set at 0.05 (5%).

## 3. Results

### 3.1. Demographic Data

Most of the respondents (26.3%) were between 31–40 years old. Adolescents and seniors were the least numerous (Table 1). Most of the respondents were women (69.9%). The analysis of the level of education of the respondents revealed that a slight majority of the respondents (53.3%) had higher education. Medical education was declared by almost 21% of the respondents. Most of the respondents live in a city with more than 200,000 inhabitants (37.4%) (Table 1).

### 3.2. The Belief That Transplantology Saves Lives in Various Population Groups

There were no differences between the individual age groups and areas of residence and the answer given to the question concerning the issue of whether organ donation can save another person’s life. There were, however, significant differences between educational level and the answer given to that question (*p* = 0.001). The number of respondents who stated that *the donation of tissues and organs can save the life* was significantly higher in the group with higher level of education than in groups with primary (*p* = 0.008), secondary (*p* = 0.003) and vocational (*p* = 0.009) education (Figure 1).

People with a non-medical type of education constitute 79.1% of the respondents. Most of the respondents declared higher educational level (53.3%), followed by vocational (32.8%), Secondary (10.1%) and Primary (3.8%). There were no differences between age range and the statement that tissue and organ donation can save life and no differences between sexes (level of agreement on the life-saving potential of tissues and organs was 93% in males and 94.4% in females). No differences were found between a rural and urban area of living (93.1% of countryside respondents and 93.9% of respondents from cities >200,000 citizens were convicted of its life-saving potential).

### 3.3. A Will to Become a Donor in Various Population Groups

The correlation between conviction about the life-saving potential of tissues and organs and readiness to become a donor was found (r_s_ = 0.34, *p* < 0.05). There was no correlation between knowing a donor and willingness to become a donor. In most age groups, only 6% of respondents know a donor (family member or friend), while this number is slightly higher (13%) only in the 51–60-years-old group. The highest percentage of respondents who know a donor live in small cities (around 50,000 citizens), but this difference is not statistically significant.

There were significant differences between the statement of agreement to become a donor between different age ranges (*p* = 0.005 for readiness to become an organ and tissue donor after death, *p* < 0.000 for readiness during lifetime), levels of education (*p* < 0.000 for readiness to become an organ and tissue donor both after death and during lifetime), medical and non-medical type of education (*p* = 0.008 for readiness to become an organ and tissue donor after death, *p* < 0.000 for readiness during lifetime) and sex (only for readiness to become an organ and tissue donor during lifetime, *p* = 0.004). Place of residence did not significantly affect the frequency of individual answers to this question (*p* = 0.172 for readiness to become an organ and tissue donor after death, *p* = 0.700 for readiness during lifetime) (Figure 2).

There is also a correlation between willingness to become a donor after death and being a donor during their lifetime (r_s_ = 0.25, *p* < 0.05).

Analysis of the data revealed that religion is one of the factors that has an impact on readiness to become a donor, both after death (*p* < 0.000) and during lifetime (*p* = 0.031) and willingness to receive a transplant, if necessary (*p* < 0.000), however, 56.90% respondents declared that religion has no impact on their decision on organ and tissue donation. The highest frequency of declaration that religion forbids to donate was observed in subjects who are not ready to be donors after death (11.11%) and during lifetime (6.00%) as well as subjects who are not ready to receive a transplant (12.00%) (Figure 3).

### 3.4. An Issue Particular Tissue/Organ Donation Agreement

Most of the respondents (79.1%) declared their consent to donate all of the organs mentioned in the question. Over 1/10 of the respondents (10.6%) expressed a reluctance to donate any organs. The obtained results revealed that 44.1% of the respondents considered that to donate after their death, according to Polish law, they must give their written consent to donate tissues and organs.

#### Skin Donation Agreement

The decision on skin donation was significantly dependent on age (*p* < 0.000), education level (*p* < 0.000), type of education (*p* < 0.000), sex (*p* = 0.040), agreement of becoming a donor, both after death as well as during lifetime (*p* < 0.000), opinion on the role of organ donation in saving lives (*p* < 0.000) and willingness to receive a transplant if necessary (*p* < 0.000) (Figure 4). Place of residence did not significantly affect the frequency of individual answers to this question (*p* = 0.108). The frequencies of answers on the question of decision on skin donation, in relation to age, sex, education and other parameters are presented on Figure 4.

The highest number of answers that skin obtained after death is desecration of a corpse was in the age groups 16–20 years old (13%), 61–70 years old (13%) and over 70 years old (14%). 53% of respondents between 16–20 years old would act according to the wishes of a deceased family member. The second highest agreement on acting accordingly to the donor’s wishes was noted in the 21–30 age group (49% of respondents). In the group of men, 9% stated that skin obtained in this way is desecration of a corpse. In the women, a smaller percentage had such a point of view. Most respondents declaring that skin donation is a desecration had vocational (20%) or primary (16%) education level. Only 4% declared higher education. Most people with higher education declared that they would follow the decision of the deceased person (44%) or declared that they would agree on skin donation from a family member because it is lifesaving (43%). Most of the respondents declaring that skin donation is desecration of a corpse were from cities of around 50,000 citizens (14%). None of the respondents with medical education thought that obtaining skin from corpses is desecration, whereas in the non-medical group, such a belief was declared by 8% of respondents. Most respondents declaring that skin donation is desecration do not think that organ/tissue donation can be lifesaving (32%) or do not know if it can be lifesaving. Those respondents also mainly do not want to be donors after death (37%) nor during their lifetime (71%). Most respondents who stated that organ/tissue donation can be lifesaving would act according to a deceased person‘s wishes (40%) (Figure 4).

### 3.5. Knowledge on Technical Aspects of Donation

About 35% of respondents answered that opt-out system is consent by means of not expressing an objection during their lifetime, and 20.6% of them were in favor of registering as a potential donor in the Central Donor Register. Due to the Polish law, if someone does not object to the Central Donor Register, theoretically organs can be harvested, but usually in Poland the opinion of the family of the deceased is respected. So, it can be said that this is a rather fictitious registry.

Most of respondents (64.9%) believe that organs and tissues are retrieved after the statement of a permanent irreversible cessation of brain activity by a team appointed specifically for this circumstance. Another group of people (24.7%) believe that donation takes place after the diagnosis of brain death based on two EEG tests (electroencephalography). Less than 7% of the study participants believe that the transplant procedure can be started after finding that cardio-respiratory action has been stopped for a period longer than 45 min. On the other hand, the remainder of the people (3.5%) claim that an unsuccessful resuscitation action lasting at least 60 min is a sufficient criterion for the collection of organs and tissues.

89% of the respondents believe that the statements of permanent, irreversible cessation of brain activities are made by specialists in the fields of medicine: anesthesiology and intensive therapy, neurology, neurosurgery and forensic medicine, which confirm the above-mentioned statement. The other answers received similar response rates.

5.4% of the respondents supported the statement that the attending physician, based on the physical examination and the history of the disease, confirmed the cessation of brain activity. Slightly less (3.9% of the respondents) believe that such a statement is made by the transplant coordinator, and the rest (1.7%) saying that a court having jurisdiction over the potential donor’s place of residence, based on expert opinions, is responsible for this decision.

Only place of residence (*p* = 0.017) and level of education (*p* = 0.030) had an impact on knowledge about the cosmetics of a corpse after skin harvesting. Most of the respondents who report that no cosmetic activity is performed are from cities of around 50,000 inhabitants (27%) (Figure 5).

Age range (*p* < 0.000), sex (*p* = 0.004), place of residence (*p* = 0.004), level of education (*p* < 0.000), type of education (*p* = 0.002) agreement of becoming a donor, both after death as well as during lifetime (*p* < 0.000), opinion on the role of organ donation in saving lives (*p* < 0.000) and willingness to receive a transplant if necessary (*p* < 0.000) had an impact on the act of talking with the family about organ/tissue donation during the respondent’s life (Figure 5).

The lowest frequency of responses declaring a conversation with the family about organ donation was in the group of the youngest, aged 16–20 years. As far as gender is concerned, the conversation on this topic was conducted significantly more often by women than men. In the case of the place of residence, such an interview was most often conducted by people living in cities with a population of over 200,000 (59.63%), and the least frequently by people from cities with a population of around 50,000 (40.19%). Most often they were people with higher education, the least often people with primary education. People with medical rather than non-medical education, as well as people who gave their informed consent to be a donor, both after death and during their lifetime, discussed organ donation more often with their families. Such an interview was conducted more often by people declaring that transplantation saves lives, and respondents who agreed to receive a transplant if necessary (Figure 6).

## 4. Discussion

The aim of this study was to examine the state of knowledge and attitudes represented by Polish society regarding the transplantation of tissues and organs, with particular emphasis on allogeneic human skin. In this study, the author’s own survey was used as a research method. This is a recognized method of public opinion polling [13].

The development of medicine contributes to increasing support and acceptance of transplantology as a commonly used form of treatment. The level of society’s knowledge that affects the broadly understood consciousness is related to many factors. They can be, among others age, gender, place of residence, level of education and its nature (e.g., medical education). Additionally, it is possible to mention, for example, contact with the donor’s family, media messages, the position of the Church, social and ethical factors, personal experiences, etc.

The Public Opinion Research Center (in Polish abbreviation- CBOS) has been studying Poles’ attitudes towards organ donation and transplantation for over twenty years. The reports published by CBOS gathered information on the respondents’ knowledge of the legal regulations concerning organ donation for transplantation, i.e., the level of awareness of the fact that in Poland, indirect consent (opt-out) is applied, not direct consent (opt-in) [14]. According to the CBOS data, the level of knowledge about the current legal status of transplantation is 10–20% (research results from 2005–2016), which in comparison with the results presented in this study is at a slightly lower level.

The results obtained in this research reveal that both the education level and type of education have a statistically significant effect on the conviction about the life-saving potential of tissue and organs. Among the group of educated people, especially in the field of medical sciences, the belief in the value of transplantology was significantly higher than in the case of people with a lower educational status. This suggests that the level of education and its type may have an impact on the level of knowledge and attitudes towards the idea of transplantology. Therefore, it can be concluded that the implementation of the educational campaign may increase the level of knowledge among people with lower education. In such a campaign, the life-saving potential of tissues and organs should be described in simple, non-medical terms as the target group of donation-promoting programs has mostly non-medical vocational education. At this point, it can be additionally explained what vocational education is. It is a type of education that occurs, among others, in Poland and represents the level of education between primary and secondary education. There is education in a specific profession (e.g., locksmith, tile maker, etc.).

Age was found to have a statistically insignificant impact on the conviction that tissue and organs can belief saving, however in the group of 16–20-year-olds, the percentage of people refusing, and hesitating was highest (12.5%) and even higher than in the group of 61–70-year-olds (10.5%) and those over 70 years old (8.6%). This proves that the education on this topic in schools is insufficient, especially in basic education, while people in vocational education are especially vulnerable to the lack of proper education in this field. Additionally, this fact can be explained by the general immaturity in the youngest group of respondents.

The results obtained revealed that sex and place of living do not have an impact on this factor. It is crucial to increase the conviction of this life-saving potential as we have proven there is a correlation between this parameter and the willingness to donate organs. It is even more important than knowing the donor. All the studied factors had an impact on the willingness to become a donor (age, sex, education level and type, place of residence, the fact of being hospitalized in our burn unit and religious beliefs). Surprisingly, the lowest readiness to become a donor after death is found in the youngest (16–20 years old) group (69%) and is even lower for becoming a donor during their lifetime (25%) which again shows the importance of transplantation promotion during basic education to avoid spreading misconceptions about what donation during their lifetime and after death looks like. The respondents from the countryside tend to declare a lower level of readiness to become a donor than those from cities, but there was no correlation between the level of education and place of living. In the case of being a donor during their lifetime, the respondents from the countryside and cities of around 200,000 citizens generally refused to become a donor.

It is hard to say whether this is related to trauma or a lack of proper knowledge administration by nurses and physicians in our burn unit. This topic needs to be more broadly examined.

The results obtained in the study are reflected in the works of other researchers. It should be emphasized once again that in our own research, approx. 94% of respondents believed the idea of donation is correct, and that donation may save someone’s life. Analogous results were obtained in the study by Roman where, again, 94% of the respondents answered the above question positively [15]. It is puzzling that the acceptance of a transplant is much higher than the consent to donate one’s organs after death. It is reasonable to propose that the reason for this phenomenon is the fear of misdiagnosis of death or the lack of understanding that brain death is irreversible and defines the moment of human death [16]. A much simpler and more prosaic explanation can also be proposed. Generally people usually prefer to gain than to lose. In that meaning the obtaining an organ/tissue would be more appreciated than losing it.

The problem related to the lack of understanding of the definition of death is a topic discussed in the works of other authors. In research presented by Kliś et al., it can be seen that 50.4% of respondents believe that death occurs only when the brain stops working and the heartbeat stops [17]. In a study by Roman [15], 38.5% of the respondents answered the same way, and in the case of Skrzypiec [18], as much as 69.3%, while in the CBOS study [14], 49% of the respondents answered this way. The results of this study are slightly different. Only 10.4% of the respondents share this opinion, while as many as 89.6% of the respondents believe that death can be concluded after the brain stops working, despite a beating heart. In the Kliś study [17], 43.3% of the respondents considered brain death to be the moment of death. In the studies by Roman [15] 55.5%, in the CBOS results [14] 49%, and in the Skrzypiec [18] studies, 18.6% of people considered brain death a sufficient criterion of human death. Such results suggest that the level of knowledge and awareness of the public about the definition of brain death is insufficient. In this approach, it is possible to point to the important role of educational programs and campaigns raising public awareness.

A lack of knowledge of the basic assumptions of Polish law among the respondents was also revealed in the answers given to the question on how to express their consent to becoming a donor after death. This study deliberately proposes a hint that reads as follows: “register as a potential donor in the Central Donor Register”, but the registration Centre mentioned in the question is a non-existent, fictitious institution. On the other hand, even though in the research presented by CBOS, the third hint was “I do not know what legal provisions are in force in Poland”, it can be concluded that in both studies a similar tendency is observed. Most of the respondents consider the “declaration of will” to be a legal form in force in Poland. Ignorance of the law in the field of transplantation may have a real reflection in the number of refusals that take place in hospitals, when families are asked for consent to donate organs and tissues of their loved ones.

There are indications that the family’s ignorance of the deceased’s wishes has a significant impact on the final decision to donate. The analysis of Góra’s research (2015) shows that 66% of the respondents talked to their relatives about transplantation [19]. Relatively similar results were obtained in this study. More than half of the respondents (52.4%) spent time talking about transplantation. However, only 25% of respondents according to CBOS (2016) confirmed such an interview [14]. In Góra’s research [19], 34% of people did not talk about the above subject with their family at all, while in their own research, 31.4% of respondents claim that they have never talked about it with their relatives. CBOS (2016) states that as many as 75% of respondents did not discuss such topics with their relatives [13]. In this research, the respondents did not remember such a fact, but 16.2% of the respondents did not exclude it.

The problem of the deficit of allogeneic skin donors is rarely discussed in the literature. While researchers have addressed the need for organ donation, few literature sources point out that human allogeneic skin graft reserves are not sufficient to meet clinical needs.

The growing deficit of allogeneic skin substitutes may be influenced by the low level of respondents’ knowledge about the cosmetic activities performed by harvesting teams after organ and tissue collection [2]. 16.1% of respondents believed that the body must be cremated after the procurement. The fact that the transplantation teams do not perform any cosmetic activities was reported by 174 people, which is 17.4% of the respondents. Transplants of this type are an important alternative to skin transplants of patients [1].

Most of the respondents declaring that skin donation is a desecration had a vocational (20%) or primary (16%) education level. Only 4% declared higher education. This type of data may indicate that despite the lack of a well-thought-out educational campaign to date, most respondents know there is an obligation to apply proper cosmetics to the body of the deceased donor after donation. Importantly, most respondents with higher education declared that they would follow the decision of a deceased person (44%), and on the other hand, respondents in that group declared that they would agree to skin donation from a family member because it is lifesaving (43%). These data can be seen as optimistic as they show the positive attitude of highly educated people towards the positive idea of transplantology. However, there is still a need for education dedicated to specific groups of society.

Most respondents declaring that skin donation is desecration of a corpse were from cities of around 50,000 inhabitants (14%). The number of people declaring that they would act according to the wishes of a deceased family member was only below 40%in this group. None of the respondents with a medical education thought that skin harvesting was desecration of a corpse, while in the non-medical group, such belief was declared by 8% of the respondents. Most of the respondents declaring that skin donation is desecration do not think that organ/tissue donation can be lifesaving (32%) or do not know if it can be lifesaving. Also, those respondents mainly do not want to be donors after death (37%) nor during their lifetime (71%). Most respondents who stated that organ/tissue donation can be lifesaving would act according to a deceased person’s wishes (40%). Surprisingly, 40% of people who declared that human skin transplantation can save life also think that organ/tissue donation is not lifesaving. Such results may suggest that the respondents do not have adequate knowledge about the effects of treatment with transplanted tissues and organs. The solution, as has already been suggested several times, could be professionally organized educational programs and social campaigns.

In that respect, it should be remarked that only age and area of living had an impact on knowledge about corpse cosmetics after skin procurement. 75% of the respondents between the ages of 21–30 declared that the normal look is restored after skin donation, whereas only 51% over 70 years old declared that cosmetic procedures are performed and 31% stated that the body needs to be cremated. Most of the respondents who think that no cosmetics is performed are from cities of 50,000 citizens (27%).

Over 60% of respondents with a medical education have already talked with their families about donation. Most of the respondents who do not recall talking with their family about organ/tissue donation also do not know if tissue/organ transplantation can be lifesaving (60%). 54% of the respondents who have talked with their family think that organ/tissue transplantation can be lifesaving. 59% of the respondents who have talked with their families want to become a donor after death and 63% are willing to become a donor during their lifetime. 56% of the respondents who have not had such a conversation with any family member do not want to be a donor after death and 46% do not want to become a donor during their lifetime. 57% of respondents who have already talked with their families want to receive a transplant if needed. 54% who have not had this talk do not want to receive a transplant if needed. Of course, it is difficult to take seriously the results of the survey in which the respondent replied that he or she would not accept a transplant in a life-threatening situation. It is highly likely that if he or she had been in this situation, their answer could have been quite different. However, this does not change the fact that providing such an answer may be a sign of extreme ignorance, which, in turn, indicates an urgent need to implement an educational program. In that special approach, it could be added that considering the statistical data, it is more likely that we will need a transplant than that we will become donors ourselves [16].

Many authors have emphasized that the allogeneic human skin graft materials collected from a deceased donor play a significant role in the therapy of heavily (extensive and deeply) burned casualties of a mass catastrophe [20,21]. It can be pointed out that human allogeneic skin graft materials may be a source of the collagenous biomaterials that are produced and evaluated in terms of tissue engineering. Such products include acellular dermal matrix ADM, which could serve as the scaffold for in vitro cultured specific cells (e.g., autologous keratinocytes, fibroblasts or allogeneic mesenchymal stem cells (MSCs)). In that manner, one could obtain a viable skin substitute as an alternative to autologous skin graft materials. This would be especially advantageous in the treatment of severely burned patients [1,2,22,23,24]. To emphasize the important role of human allogeneic skin transplants in the treatment of burn patients, reference can be made to the guidelines of the European Burn Association (EBA). One of the EBA recommendations is that any burn treatment unit or ward should have access to a skin bank [25]. In accordance with that approach, skin banks can be considered one of the key institutions in the treatment of severely burned patients. As such, it may be accepted that the main requirement for a skin bank is to provide the required amount of allogeneic human skin graft materials for normal, everyday clinical routine work as well in the case of a mass disaster.

In this approach, it could be suggested that there is a specific dissonance between the high role of allogeneic skin transplants in the treatment of burns (as life-saving grafts) and the social acceptance of this material, confirmed by this research in the confrontation with the actual number of skin donations procured in Poland. The indication of this issue can be considered an important contribution of this work to the social and clinical significance of the problem. However, the issue of the insufficient number of allogeneic skin donations remains unsolved. In that perspective further consistent research based on increasingly larger groups of participants is required. That new study would include the medical personnel directly involved in the procedures of coordination and authorization of organ and tissue donations.

An aspect of the work that may raise some doubts is the difficulty in obtaining the representativeness of the results. Despite it could be agreed that the study performed involved the sumptuous and diverse study group. It cannot be concluded that the surveyed community fully corresponds to the cross-section of Polish society. It could probably result from the fact that certain groups were more willing to participate in the study, which was fully voluntary (which should be emphasized).

There is no doubt that tissue and organ transplantation is one of the most important achievements of modern medicine. The results obtained in this study confirm this thesis. The respondents showed great understanding and acceptance of the idea of saving human life by transplanting tissues and organs. However, an in-depth analysis showed that despite a positive attitude, the respondents lacked basic knowledge about the issue under study. Therefore, it may be necessary to take measures to counteract false attitudes towards organ and tissue transplantation.

## 5. Conclusions

In accordance with the data presented it might be concluded that the respondents has a positive attitude towards transplantology. However, the level of knowledge in that field remains unsatisfactory. Thus, well planned educational schemes and awareness raising campaigns are required.

## Figures and Tables

**Figure 1 ijerph-19-13875-f001:**
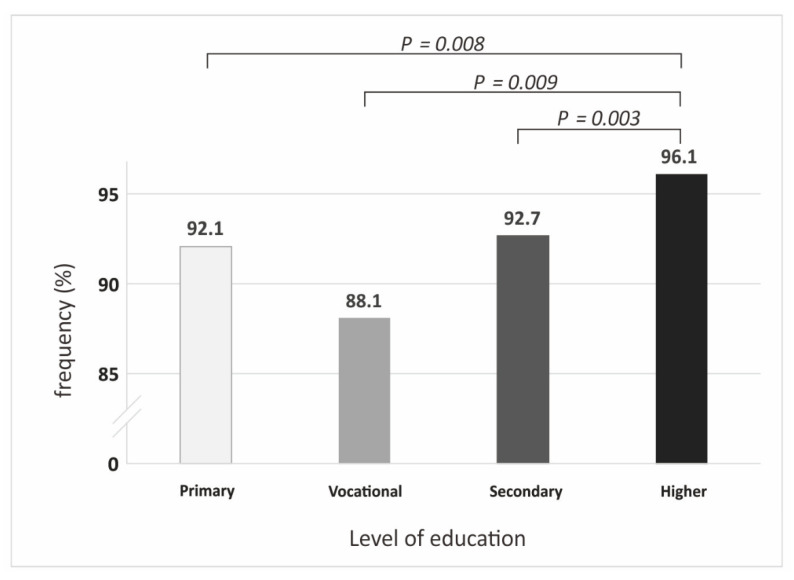
The frequency of answers given to the question of whether organ donation can save another person’s life by education level.

**Figure 2 ijerph-19-13875-f002:**
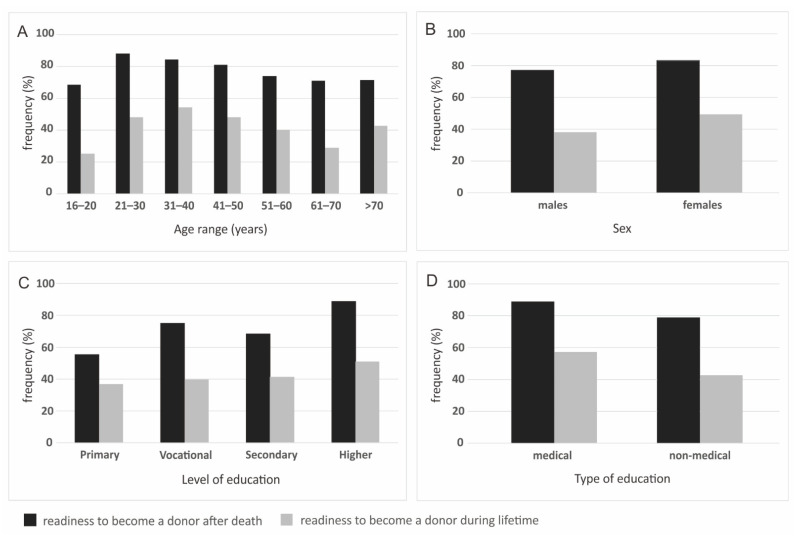
Readiness to become an organ and tissue donor after death and during their lifetime, by: (**A**) age range; (**B**) sex; (**C**) level of education and (**D**) type of education.

**Figure 3 ijerph-19-13875-f003:**
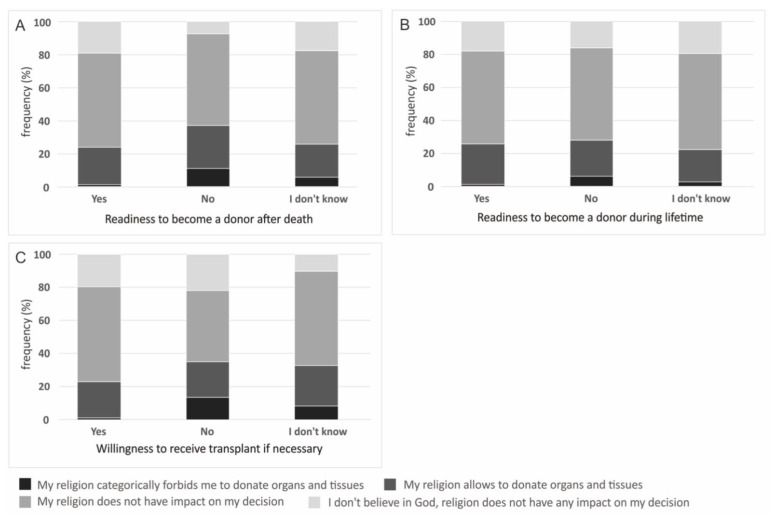
Impact of religion on: (**A**) readiness to become a donor after death; (**B**) readiness to become a donor during lifetime and (**C**) willingness to receive transplant if necessary.

**Figure 4 ijerph-19-13875-f004:**
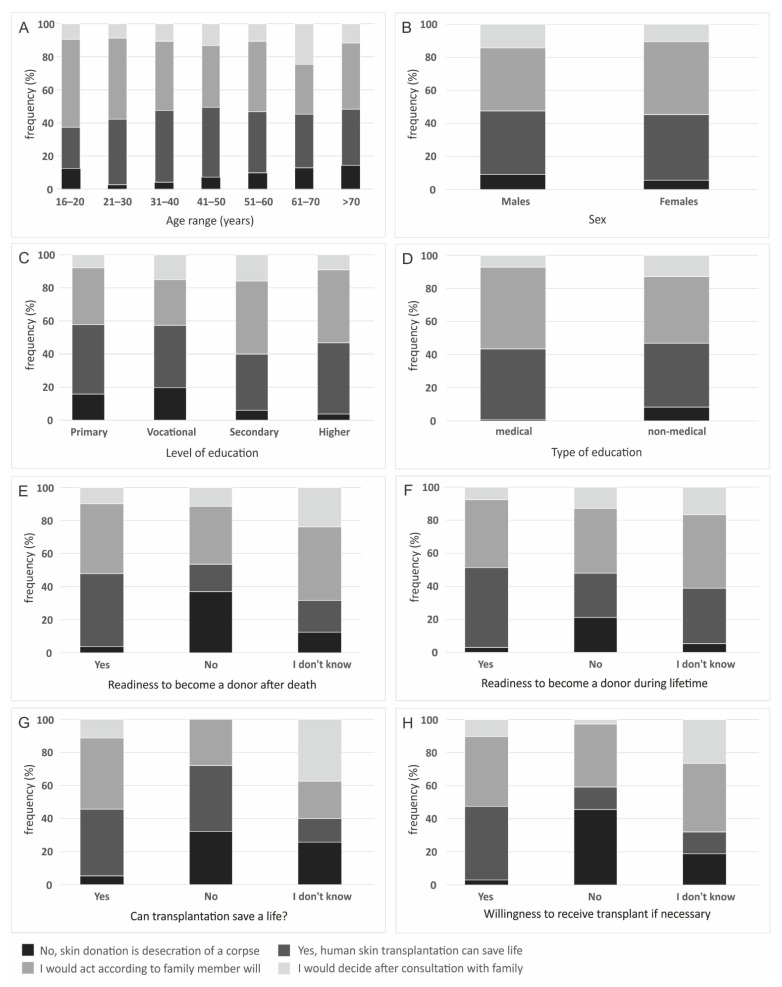
Decision on skin donation, by: (**A**) age range; (**B**) sex; (**C**) level of education; (**D**) type of education; (**E**) readiness to become a donor after death; (**F**) readiness to become a donor during lifetime; (**G**) opinion on the role of organ donation in saving lives and (**H**) willingness to receive transplant if necessary.

**Figure 5 ijerph-19-13875-f005:**
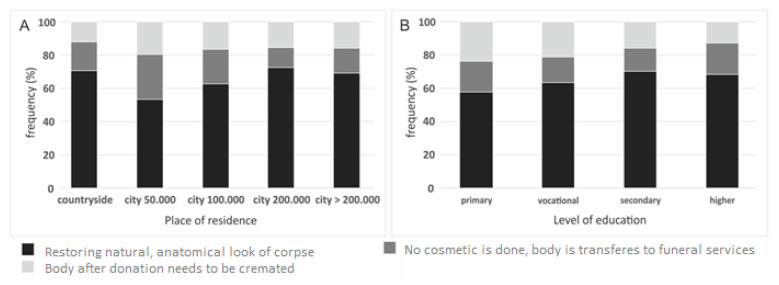
Corpse cosmetics after skin donation, by: (**A**) place of residence; (**B**) level of education.

**Figure 6 ijerph-19-13875-f006:**
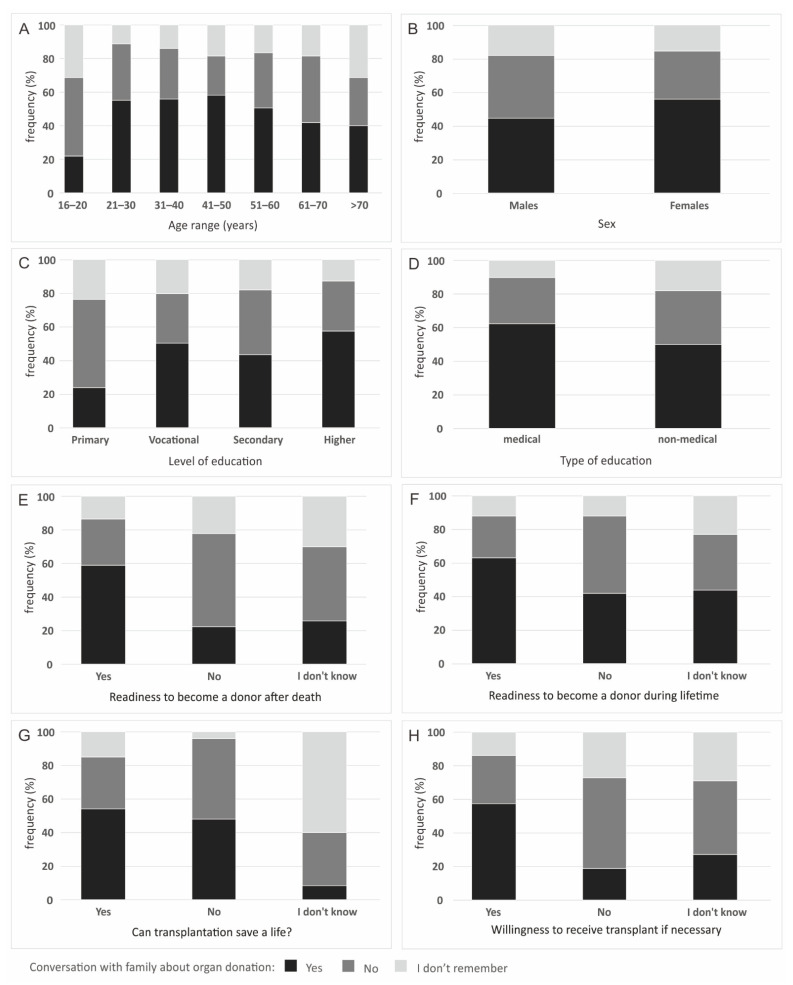
Conversation with family about organ donation, by: (**A**) age range; (**B**) sex; (**C**) level of education; (**D**) type of education; (**E**) readiness to become a donor after death; (**F**) readiness to become a donor during lifetime; (**G**) opinion on the role of organ donation in saving lives and (**H**) willingness to receive transplant if necessary.

**Table 1 ijerph-19-13875-t001:** Characteristics of survey participants.

Demographic Parameter		*n* (%)
Age range (years)	16–20	32 (3.20)
	21–30	232 (23.20)
	31–40	263 (26.30)
	41–50	190 (19.00)
	51–60	162 (16.20)
	61–70	86 (8.60)
	<70	35 (3.50)
Sex	female	699 (69.90)
	male	301 (30.10)
Place of residence	countryside	58 (5.80)
	city with about 50,000 inhabitants	107 (10.70)
	city with about 100,000 inhabitants	274 (27.40)
	city with about 200,000 inhabitants	187 (18.70)
	city with above 200,000 inhabitants	374 (37.40)
Education level	primary	38 (3.80)
	vocational	328 (32.80)
	secondary	101 (10.10)
	higher	533 (53.30)
Type of education	medical	209 (20.90)
	non-medical	791 (79.10)
Hospitalised in Burn Treatment Center	yes	144 (14.40)
	no	856 (85.60)

## Data Availability

All data generated or analyzed during this study are included in this article. Further enquiries can be directed to the corresponding author.

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
