# Peer review of "A Survey of 1000 Respondents on the Polish Population’s Knowledge and Attitudes about Tissue/Organ Donation and Transplantation in Times of Allogeneic Tissue Shortage"

_ijerph, 2022, doi:10.3390/ijerph192113875_

Round 1
Reviewer 1 Report
This is an interesting article about an important topic. The study consists of 23 questions on public knowledge and attitudes towards organ and tissue donation and transplantation in Poland. My main concern is about the representativeness of the results. From the Materials and Methods section, it is unclear how and when the survey was conducted and to whom it was distributed.
I have no major concerns about the overall quality of this study but some improvements could be made. In the following, I will comment sequentially on the issues I identified.
TITLE
The title does not adequately reflect the content of the article. It mentions the population factors that influence the number of skin donors but, in my opinion, this is not what this study actually does. The study explores the knowledge and attitudes of the Polish population with regard to organ/tissue donation/transplantation, including their willingness to donate. Then, it analyses these data in the light of sociodemographic factors. However, it does not provide evidence as to which, whether and to what extent the knowledge, attitudes, and sociodemographic factors influence the number (rate?) of skin donors in Poland. I would recommend in the title a clearer description of the study, including its methods.
ABSTRACT
In the abstract (as well as in the last sentence of the introduction), it is mentioned that the goal was to analyze the level of knowledge of Polish society. However, the description of the results also mentions attitudes. Both knowledge and attitudes should be acknowledged. Overall, I think the redaction of the abstract can be improved by focusing on the most important results and by providing specific data.
INTRODUCTION
— Lines 47-49. It is mentioned that the number of multi-organ donations (529) in Poland is ten times higher than skin donations (54). First, does 529 multi-organ donations mean 529 individual donors or 529 organs procured? Second, what is the reason for this disparity? Is it because of a lack of individual consent or family authorisation for skin recovery? Please provide some context and explanation here.
— Lines 49-56. Why should the attitudes and knowledge of society and medical personnel be considered? Are you assuming that attitudes and knowledge are related to the low figures of skin recovery? If so, this should be explicited and justified. If there is a different reason, this should also be explicited and justified.
— Lines 56-58. Why are you mentioning campaigns? Are you implying that communication campaigns can have an effect on public attitudes and knowledge (which could in turn influence the numbers/rates of skin recovery)? Again, your assumptions should be explicited and justified.
MATERIAL AND METHODS
— L 69-74. The sampling and recruitment methods must be described. Basic information is missing: When, Where, Who and How? When was the study conducted? Where was the printed survey disseminated? Who was targeted and approached (and why), both online and offline, and how were they selected and approached? What bias checks have you conducted? All this information is crucial to assessing the methodological quality of the study, let alone its representativeness.
— Statistical analysis. I am not competent to comment on this subsection.
— Questionnaire (supplementary material). Extensive editing of English language is required. For example, Q7 mentions the "download" (sic) of tissues and organs.
RESULTS
— The first paragraph mentions 70% of respondents are women and 21% have medical education. This raises the question of the representativeness of this study and of potential sources of bias. Indeed, both figures seem to be higher than the average in the Polish population. Therefore, the authors should discuss this in the discussion, for example in a paragraph or subsection on the limitations of this study.
— Table 1 (and in the text). I do not understand what "vocational" means as an education level. Please provide some clarification.
— L 192. Typo: "that adopt-out system"
— Figure 5. Typo: "to funreal services"
— Overall, the results section could be improved by displaying in a more structured way the different kinds of results, and by avoiding lengthy descriptions of data that are shown in the Figures.
DISCUSSION
— L 306-309. There an obvious alternative reason: people usually prefer gains to loses, e.g. obtaining an organ/tissue rather than losing it. Receiving an organ or tissue is directly and highly beneficial for recipients, while donating an organ or tissue is not beneficial and can even be perceived as harmful by donors.
— L 308. I think the word "Dilemma" is not the most appropriate here.
— L 421. Typo: "canbe"
— Overall, the discussion could be improved by avoiding unnecessary repetitions of the data already provided in the results section. I highly recommend a shorter and crisper redaction of the discussion section.
CONCLUSIONS
— Typo: "Conclussions"
— The sentence "Polish society represents a positive attitude…" should be revised. Do you mean "has a positive attitude"?
— As it stands, the conclusion does not make much sense. It does not reflect the content of the article or its main findings.
— Overall, you cannot draw conclusions about Polish society in general, in terms of knowledge and attitudes, because you have not shown that your study is representative of the Polish population.
Author Response
Dear Reviewer,
We would like to thank the reviewers for their insightful and valuable comments. We tried to take into account all of them and to correct the manuscript according to the recommendations. We hope that the corrections will improve the quality of the manuscript. We incorporated all the changes and marked them in the manuscript. Reviewers opinions will significantly improve this work, we have expanded the literature of the manuscript with 11 additional publications. The changed fragments in the text are underlined in yellow. The manuscript has been corrected by a native speaker.
Comments of Reviewer 1 :
Comment 1 :
The title does not adequately reflect the content of the article. It mentions the population factors that influence the number of skin donors but, in my opinion, this is not what this study actually does. The study explores the knowledge and attitudes of the Polish population with regard to organ/tissue donation/transplantation, including their willingness to donate. Then, it analyses these data in the light of sociodemographic factors. However, it does not provide evidence as to which, whether and to what extent the knowledge, attitudes, and sociodemographic factors influence the number (rate?) of skin donors in Poland. I would recommend in the title a clearer description of the study, including its methods.
Answear to Comment 1 :
Thank you for this valuable comment. We changed title according to your suggestions.
Title has been changed and now it is: “1000 respondents survey on the knowledge and attitudes of the Polish population with the regard on tissue/organ donation and transplantation in a view of the deficit of allogeneic skin graftsin Poland”.
Comment 2 :
ABSTRACT
In the abstract (as well as in the last sentence of the introduction), it is mentioned that the goal was to analyze the level of knowledge of Polish society. However, the description of the results also mentions attitudes. Both knowledge and attitudes should be acknowledged. Overall, I think the redaction of the abstract can be improved by focusing on the most important results and by providing specific data.
Answear to comment 2 :
Thank you very much for this important comment. Reviewer 1 is right, abstract must be improved. We had corrected abstract according to te Reviewer 1 suggestions.
Abstract has been rearanged.
Abstract: Tissue-engineered human allogeneic skin grafts retrieved from a deceased donor play an important role in the therapy of extensive and deeply burned patients. However, there is a vital deficit of allogeneic skin donors, and the reserves of human allogeneic skin grafts are not sufficient. The goal of this work was to analyze the level of knowledge and attitudes of Polish society in the field of transplantation, with particular emphasis on allogeneic skin transplantation. The study used a self-made questionnaire comprised of 23 questions. 1000 respondents took part in this research. The respondents were a diverse group in terms of age, sex, education, and place of residence. The obtained results show a general positive attitude of the respondents towards the idea of transplantology. However, people with lower education presented a more negative attitude towards the donation of tissues and organs. Additionally younger people were not able to clearly declare readiness for organ procurement. What is more data analysis revealed certain gaps in more detailed knowledge and surprising attitudes. In that respect, the lack of awareness about the criteria for determining brain death could be mentioned. There was also a lack of acceptance for skin procurement in specific population groups. It can therefore be concluded that a key role in the success of the idea of transplantation in Poland is the broad and systematic education of the society.
All changes in the main text are underline in yellow.
Comment 3 :
INTRODUCTION
— Lines 47-49. It is mentioned that the number of multi-organ donations (529) in Poland is ten times higher than skin donations (54). First, does 529 multi-organ donations mean 529 individual donors or 529 organs procured? Second, what is the reason for this disparity? Is it because of a lack of individual consent or family authorisation for skin recovery? Please provide some context and explanation here.
Answear to Comment 3 :
Thank you very much for this valuable suggestions. We changed the text according to Reviewer 1 suggestions. There is no doubt that your suggestion will improve the quality of this manuscript. Our explanation is below. In main manuscript changed text is underlined in yellow.
„…38 million citizens, 529 multi-organ donations (real deceased donors) took place in 2020, while only 54 skin donations took place [7]. The likely cause of this unfavorable phenomenon may be the fact that in Poland there are only two centers for the procurement, preparation and distribution of human allogeneic skin grafts. There are therefore two procurement teams that collect allogeneic skin and, what is more, their range is locally limited. In this approach, there are proposals for systemic solutions which could lead to a central bank of allogeneic skin establishment in the event of a mass disaster (including war) [2]. However, at present, a frequently observed situation is retrieval of tissues / organs from a deceased donor during which, for example, corneas are retrieved and the skin is not. It should be remarked that the number of corneal retrievals was 744 while skin donation was 54 in 2020 [2, 7]. The high number of corneal retrievals may be due to the fact that corneas are pocured by more centers and additionally in forensic and funeral homes. Therefore, paradoxically, the greater number of donations of this type may result in greater awareness of the role of corneal transplants as well as the course of the donation itself, which translates into a greater number of donations. Nevertheless, there is some kind of concern among the public, and possibly the medical staff, regarding the skin procurement. [data not shown].” Comment 4 :
— Lines 49-56. Why should the attitudes and knowledge of society and medical personnel be considered? Are you assuming that attitudes and knowledge are related to the low figures of skin recovery? If so, this should be explicited and justified. If there is a different reason, this should also be explicited and justified.
Answear to Comment 4 :
Reviewer 1 is right, there is a need of explanation. Thank you very much for this comment. We think that explanation was attached in the text above. Your suggestions will definitely improve this manuscript, thank
Comment 5 :
— Lines 56-58. Why are you mentioning campaigns? Are you implying that communication campaigns can have an effect on public attitudes and knowledge (which could in turn influence the numbers/rates of skin recovery)? Again, your assumptions should be explicited and justified.
Answear to Comment 5 :
Thank you very much to Reviewer 1 for this comment. It has been proved that education sessions are able to addressed myths about transplantation and help to understand its purposes [12]. Therefore, an increase in the number of donations can be expected. However it should be remarked.
This suggestions will improve this manuscript.
Commment 6 :
MATERIAL AND METHODS
— L 69-74. The sampling and recruitment methods must be described. Basic information is missing: When, Where, Who and How? When was the study conducted? Where was the printed survey disseminated? Who was targeted and approached (and why), both online and offline, and how were they selected and approached? What bias checks have you conducted? All this information is crucial to assessing the methodological quality of the study, let alone its representativeness.
Answear to Comment 6 :
Thank you for this valuable comment. Reviewer 1 is right. We changed the text in these sections according to your comments. Changed text is below, in the main document is underlined in yellow as well.
The study consisted of 23 closed questions and the survey was anonymous. The study lasted from January 4, 2021 to March 1, 2021 to achieve a response rate of 1000 surveys. Both a printed survey and an electronic survey generated by the Google form were used. In the case of the electronic version of the survey, the available online tools for social media (e.g. Facebook, LinkedIn) were used to disseminate the survey. The questionnaire was distributed to the network of contacts, but no form of inducement or coercion was used to participate in the study. The recruitment of respondents to the study consisted of voluntary consent to participate in the survey. In case of printed version of the questionnaire participants were recruited among nursing students, CLO patients, CLO surgical clinic patients, CLO medical and administrative staff, employees of selected industrial plants, family members of researchers.
Comments of Reviewer 1 :
— Statistical analysis. I am not competent to comment on this subsection.
— Questionnaire (supplementary material). Extensive editing of English language is required. For example, Q7 mentions the "download" (sic) of tissues and organs.
Our answear:
Thank you for mentioning this informations. We already corrected the language and text according to these suggestions. Thank you very much for this comment.
Comment 7 :
RESULTS
— The first paragraph mentions 70% of respondents are women and 21% have medical education. This raises the question of the representativeness of this study and of potential sources of bias. Indeed, both figures seem to be higher than the average in the Polish population. Therefore, the authors should discuss this in the discussion, for example in a paragraph or subsection on the limitations of this study.
Thank you very much for pointing this important information. We changed the text according your suggestions. All changed fragments are underlined in yellow.Please see the Discussion section, text is below.
"An aspect of the work that may raise some doubts is the difficulty in obtaining the representativeness of the results. Despite it could be agreed that the study performed involved the sumptuous and diverse study group. It cannot be concluded that the surveyed community fully corresponds to the cross-section of Polish society. It could probably result from the fact that certain groups were more willing to participate in the study, which was fully voluntary (which should be emphasized)."
Minor comments/suggestions of Reviewer 1 :
Comment :
— Table 1 (and in the text). I do not understand what "vocational" means as an education level. Please provide some clarification.
Answear :
Thank you very much fot this suggestions. At this point, it can be additionally explained what vocational education is. It is a type of education that occurs, among others, in Poland and represents the level of education between primary and secondary education. There is education in a specific profession (e.g. locksmith, tile maker, etc.).
Comment :
— L 192. Typo: "that adopt-out system"
— Figure 5. Typo: "to funreal services"
Answear :
All typing mistakes have been corrected
Comment :
— Overall, the results section could be improved by displaying in a more structured way the different kinds of results, and by avoiding lengthy descriptions of data that are shown in the Figures.
Answear
Thank you very much for this valuable comment. You are right. We changed text according your suggestions. The Results section has been structured.
Comment :
DISCUSSION
— L 306-309. There an obvious alternative reason: people usually prefer gains to loses, e.g. obtaining an organ/tissue rather than losing it. Receiving an organ or tissue is directly and highly beneficial for recipients, while donating an organ or tissue is not beneficial and can even be perceived as harmful by donors.
Our answear :
…death [15]. A much simpler and more prosaic explanation can also be proposed. Generally people usually prefer to gain than to lose. In that meaning the obtaining an organ/tissue would be more appreciated than losing it.
Comment :
— L 308. I think the word "Dilemma" is not the most appropriate here.
Our answear :
Dilemma was changed for phenomenon
Comment :
— L 421. Typo: "canbe"
Our answear :
Corrected
Comment :
— Overall, the discussion could be improved by avoiding unnecessary repetitions of the data already provided in the results section. I highly recommend a shorter and crisper redaction of the discussion section.
CONCLUSIONS
— Typo: "Conclussions"
Our answear :
Thank you very much for this valuable comment. We changed the text according your suggestions. It is in current form of manuscript. It is corrected.
Comment :
— The sentence "Polish society represents a positive attitude…" should be revised. Do you mean "has a positive attitude"?
Our answear :
Corrected
Comment :
— As it stands, the conclusion does not make much sense. It does not reflect the content of the article or its main findings.
— Overall, you cannot draw conclusions about Polish society in general, in terms of knowledge and attitudes, because you have not shown that your study is representative of the Polish population.
Our answear :
Thank you very much for this comments. In accordance to the data presented it might be concluded that Polish society has a positive attitude towards transplantology. However the level of knowledge in that field remains unsatisfactory. Thus well planned educational schemes and awareness raising campaigns are required.
Thank you very much for your very important suggestions. We try to explain all concerns. Your suggestions will improve the quality of this manuscript. Thank you very much for all your work with this manuscript.
Sincerely yours,
Karol Szyluk, Marcin Gierek, Wojciech Łabuś
Reviewer 2 Report
In the introduction: Are there any studies related to your topic about organ donation, campaigns, education, etc. in Poland? (if available, give a summary of the results)
About the effectiveness of campaigns for organ donation, have these campaigns been effective that they need to be designed and implemented? Evidence should be mentioned.
Methodology: very incomplete and without details. How was the questionnaire designed? Based on which sources or studies? Its validity and reliability? Scoring? Sampling and its details? Ethics approval?
Results: The presentation of the results is not good and I was not able to communicate well or your findings.
Limitations of the study? And especially about the generalizability of the results to the whole Polish society?
Author Response
Dear Reviewer,
We would like to thank the reviewers for their insightful and valuable comments. We tried to take into account all of them and to correct the manuscript according to the recommendations. We hope that the corrections will improve the quality of the manuscript. We incorporated all the changes and marked them in the manuscript. Reviewers opinions will significantly improve this work, we have expanded the literature of the manuscript with 11 additional publications. The changed fragments in the text are underlined in yellow. The manuscript has been corrected by a native speaker.
Comment 1 :
In the introduction: Are there any studies related to your topic about organ donation, campaigns, education, etc. in Poland? (if available, give a summary of the results)
Our explanation :
Thank you very much for this comment. We changed the text according to your suggestions. There are few studies on this topic. However we express the hope that this sentence would be the answear:
…Zhang et al.[2017] and Wilczek-Rużyczka et al. [2014] proved that there is a need to provide appropriate training regarding donation to increase donation rates [9, 10]. The results of the available studies differ depending on the country and studied group. In order to perform an efficient educational and awareness raising campaign, it is crucial to know the target group, and due to the generation gap, present campaigns may be outdated [11, 12]. It has been proved that education sessions are able to addressed myths about transplantation and help to understand its purposes [12]. Therefore, an increase in the number of donations can be expected.
Comment 2:
About the effectiveness of campaigns for organ donation, have these campaigns been effective that they need to be designed and implemented? Evidence should be mentioned.
Our answear :
Thank you very much for this valuable comment.
Please see text above. We hope that this explanation will improve this manuscript.
Comment 3:
Methodology: very incomplete and without details. How was the questionnaire designed? Based on which sources or studies? Its validity and reliability? Scoring? Sampling and its details? Ethics approval?
Our answear :
Thank you very much for this suggestions. We changed the text according to your comments.
The study consisted of 23 closed questions and the survey was anonymous. The study lasted from January 4, 2021 to March 1, 2021 to achieve a response rate of 1000 surveys. Both a printed survey and an electronic survey generated by the Google form were used. In the case of the electronic version of the survey, the available online tools for social media (e.g. Facebook, LinkedIn) were used to disseminate the survey. The questionnaire was distributed to the network of contacts, but no form of inducement or coercion was used to participate in the study. The recruitment of respondents to the study consisted of voluntary consent to participate in the survey. In case of printed version
The proposed research scheme was positively opinioned by the Institutional Bioethics Committee (Institutional Review Board) - code number is in main document.
Institutional Review Board Statement: This research complies with the guidelines for human studies and was conducted ethically in accordance with the World Medical Association Declaration of Helsinki. Institutional Review Board of Center for Burns Treatment in Siemianowice Slaskie, Poland. Code of approval : 1/2019
Comment :
Results: The presentation of the results is not good and I was not able to communicate well or your findings.
Thank you very much for this valuable comment. You are right. We changed text according your comments. The Results section has been structured
Comment :
Limitations of the study? And especially about the generalizability of the results to the whole Polish society?
Thank you very much for this important information.
An aspect of the work that may raise some doubts is the difficulty in obtaining the representativeness of the results. Despite it could be agreed that the study performed involved the sumptuous and diverse study group. It cannot be concluded that the surveyed community fully corresponds to the cross-section of Polish society. It could probably result from the fact that certain groups were more willing to participate in the study, which was fully voluntary (which should be emphasized).
Thank you Reviewer 2 for your valuable comments. We hope that these changes will improve the quality of this manuscript. Thank you very much for your work with our manuscript.
Sincerely yours,
Karol Szyluk, Marcin Gierek, Wojciech Łabuś
Round 2
Reviewer 1 Report
I recommend that the authors have the text proofread to correct certain errors or clumsiness and to improve the wording of certain sentences. Also, starting the title with a number seems strange to me, but that is a matter I leave to the editor's discretion.
In the Conclusion, one cannot draw conclusions about Polish society as a whole from this study. I suggest replacing "Polish society" with "the respondents." I have no additional comments regarding the content.
Author Response
Comment 1 :
I recommend that the authors have the text proofread to correct certain errors or clumsiness and to improve the wording of certain sentences. Also, starting the title with a number seems strange to me, but that is a matter I leave to the editor's discretion.
Our explanation :
We thank Reviewer 1 for such a valuable comment. We have proofread the manuscript and corrected spelling errors. We are trying to better explain some parts of the text that may not be very well understood by a non-Polish reader. E.g. Central Donor Register - we have explained this part more thoroughly to make it clearer for readers. We changed the title of the paper according to Reviewer 1's suggestions. Title is : " A survey of 1,000 respondents on the Polish population's knowledge and attitudes about tissue/organ donation and transplantation in times of allogeneic tissue deficit."
Thank you very much for this important feedback, which will definitely improve the quality of this article.
In the Conclusion, one cannot draw conclusions about Polish society as a whole from this study. I suggest replacing "Polish society" with "the respondents." I have no additional comments regarding the content.
Reviewer 1 is absolutely right - we changed Polish society to a "respondents" as Reviewer 1 was suggesting. Thank you very much for these important comments and for your insightful work with this manuscript.
All new parts of the manuscript are in yellow in the latest version of the manuscript.